# Effect of the Tumor Suppressor miR-320a on Viability and Functionality of Human Osteosarcoma Cell Lines Compared to Primary Osteoblasts

**Laura De-Ugarte [1], Susanna Balcells [2], Robert Guerri-Fernandez [1], Daniel Grinberg [2], Adolfo Diez-Perez [1], Xavier Nogues [1] and Natalia Garcia-Giralt [1,*]**

[1] Musculoskeletal Research Group, IMIM (Hospital del Mar Medical Research Institute), Centro de Investigación Biomédica en Red en Fragilidad y Envejecimiento Saludable (CIBERFES), ISCIII, 08003 Barcelona, Spain; laura.deugarte@gmail.com (L.D.-U.); rguerri@imim.es (R.G.-F.); ADiez@parcdesalutmar.cat (A.D.-P.); xnogues@parcdesalutmar.cat (X.N.)

[2] Centro de Investigación Biomédica en Red de Enfermedades Raras (CIBERER), Department of Genetics, Microbiology and Statistics, Facultat de Biologia, Universitat de Barcelona, ISCIII, IBUB, IRSJD, 08028 Barcelona, Spain; sbalcells@ub.edu (S.B.); dgrinberg@ub.edu (D.G.)

[*] Correspondence: ngarcia@imim.es; Tel.: +34-93-3160497

**Abstract:** The miR-320a regulates a number of genes involved in various physiological processes. In particular, it has been reported as a tumor suppressor in several types of human cancers and involved in osteoporotic fracture and osteoblast function. Hence, the role of miR-320a has been evaluated in tumor cells and in primary cells in a separated context, but its effect has never been explored in a comparative manner. The present study aims to evaluate the cellular effects of miR-320a on human osteosarcoma cell lines (MG-63 and U2OS) compared to that on primary human osteoblasts (hOBs). miR-320a was either overexpressed or inhibited in all cell lines, and cell proliferation and viability were analyzed. Additionally, the effects of miR-320a on matrix mineralization, alkaline phosphatase activity, and oxidative stress were also evaluated in order to assess osteoblast functionality. In osteosarcoma cells, miR-320a overexpression reduced cell viability and proliferation, while in hOB cell viability was not affected and proliferation even was increased. The overexpression of miR-320a in both osteosarcoma cells and hOBs reduced the mineralization capacity. Finally, an increased oxidative stress was detected in all cells after miR-320a overexpression mainly in osteosarcoma. In conclusion, the overexpression of miR-320a increased stress oxidation levels, which could be involved in the reduced osteoblast performance, even though the cell viability was only affected in osteosarcoma cells.

**Keywords:** osteosarcoma; MG-63; U2OS; primary osteoblasts; miR-320a

## 1. Introduction

MicroRNAs (miRNAs) are part of cellular processes involved in bone metabolism regulation and ageing-related mechanisms [1]. Among bone-related microRNAs, miR-320a was found to be elevated in osteoporotic bone [2] and osteoarthritic cartilage tissue [3,4]. This miRNA was also detected in serums as a circulating microRNA, and it was proposed as a potential biomarker for osteonecrosis of femoral head [5]. Moreover, this miRNA is mainly involved in cancer malignancies including osteosarcoma, in which it acts as a tumor suppressor [6–11].

In bone tissue, miR-320a is expressed both in primary human osteoblasts (hOBs) and in differentiated osteoclasts (hOCs) [12], and it is associated with hOB function [13] by targeting genes involved in bone metabolism [14,15]. In a previous study by Wu et al. (2016) [16], a reduced

miR-320a expression was observed in several osteosarcoma cell lines compared to in the human normal osteoblastic cell line hFOB1.19, but the miRNA-320a effect on cell proliferation was only tested in U2OS. Moreover, no other osteoblast functional assessments were evaluated.

Hence, this miRNA has been extensively studied in osteosarcoma and primary osteoblasts in a separated manner, but it has never been tested in both cell types in the same experimental context.

In this study, in vitro cell assays were performed in order to evaluate the miR-320a effect on human osteosarcoma cells compared to that on hOBs. For this purpose, miR-320a was either overexpressed or inhibited in MG-63 and U2OS cell lines, and their cell viability, proliferation, and oxidative stress were evaluated in comparison to those of hOBs. Moreover, cell functionality by measuring matrix mineralization and alkaline phosphatase (ALP) activity was also tested.

## 2. Materials and Methods

### 2.1. Cell Culture

hOBs were obtained from knee trabecular bone after prosthesis replacement following the protocol described by De-Ugarte et al. [13]. The study was approved by the Clinical Research Ethics Committee of Parc de Salut MAR (registry numbers: 2010/3882/I and 2013/5266/I). The study was carried out in accordance with the Declaration of Helsinki, and written informed consent was obtained from all participants included in the study. Briefly, trabecular bone was dissected in small pieces, washed in phosphate-buffered solution (PBS) and placed into a 15 cm-diameter Petri dish in Dulbecco's modified Eagle's medium (DMEM) supplemented with a 10% fetal bovine serum (FBS; Merck Life Science S.L.U., Madrid, Spain), penicillin and streptomycin (100 UI/mL) (Merck Life Science S.L.U., Madrid, Spain), ascorbic acid (100 µg/mL) (Merck Life Science S.L.U., Madrid, Spain), and fungisone (0.4%) (Gibco, Fisher Scientific SL; Madrid, Spain ). Explants were incubated at 37 °C in a humidified atmosphere of 5% $CO_2$, and the medium was changed once a week until cell confluence. All experiments were performed at maximum passage 2.

In parallel, osteosarcoma cell lines (U2OS and MG-63) were cultured in the same conditions as described for hOBs.

### 2.2. Cell Transfection

To evaluate cell viability (MTS assay), cell proliferation (BrdU incorporation), ALP activity, and oxidative stress levels (CellRox® Green Reagent assay), 96-well plates with 12,000 cells/well were used. For Alizarin red assays, a 24-well plate with 45,000 cells/well was used.

The transfection protocol was described by De-Ugarte et al. [13]. Briefly, the hsa-miR-320a mirVana™ mimic at 100 nM or its inhibitor at 400 nM (Ambion® Life Technologies; Madrid, Spain) was transfected using Lipofectamine RNAiMAX (Invitrogen; Carlsbad, USA) according to manufacturer instructions. mirVana™ miRNA Mimic Negative Control and miRNA Inhibitor Negative Control were used as controls. The mature miR-320a sequence corresponded to hsa-miR-320a-3p (5′-AAAAGCTGGGTTGAGAGGGCGA-3′).

Transfection efficiency was controlled using qPCR and miRIDIAN microRNA Mimic Transfection Control with Dy547 as previously described by De-Ugarte et al. [13]. The same efficiencies were detected in all tested cell lines (data not shown).

### 2.3. miR-320a Quantification by qPCR

To evaluate the miR-320a expression levels in hOBs, U2OS, and MG-63, total RNA was extracted using the miRNeasy Mini Kit (QIAGEN Iberia SL; Madrid, Spain) according to manufacturer instructions. Then, 1 µg of total RNA was reverse-transcribed using the miScript II RT Kit (QIAGEN Iberia SL; Madrid, Spain). cDNA was assayed in 10 µL qPCR reactions in 384-well plates using MiScript SYBR Green PCR Kit according to the protocol. The mature miR-320a sequence was used as a forward primer, and the Universal primer was used as a reverse primer. Amplification was performed in the QuantStudio 12K Flex Real-Time PCR System (Applied Biosystems; Waltham, Massachusetts, USA), and the ExpressionSuite software v.1.0.3 (Life Technologies; Carlsbad,

California, USA) was used for the determination of relative quantification (RQ) by 2-$\Delta\Delta$Ct method. Global normalization was used to normalize qPCR results.

## 2.4. Cell Viability Assay

Viability was assessed 48 h after miR-320a transfection in hOBs ($n = 3$), MG-63 ($n = 3$), and U20S ($n = 3$) using the CellTiter 96® AQueous One Solution Cell Proliferation Assay (Promega; Madison, WI, USA) according to manufacturer instructions. The absorbance was measured at 490 nm with a scanning multiwell spectrophotometer Infinite M200 (Tecan).

## 2.5. Cell Proliferation Quantification

Cell Proliferation Elisa, BrdU (Colorimetric) Kit (Roche; Sant Cugat del Vallès, Spain) was used to quantify cell proliferation after 48 h of miR-320a or inhibitor transfection in hOBs ($n = 3$), MG-63 ($n = 3$), and U2OS ($n = 3$), based on the measurement of BrdU incorporation during DNA synthesis.

## 2.6. Osteoblast Mineralization Capacity

For this test, hOBs and osteosarcoma cell lines were cultured with an osteoblastic medium supplemented with 5 mM β-glycerophosphate (Merck Life Science S.L.U., Madrid, Spain); the medium was changed three times per week. Osteoblast functionality was analyzed through mineralization capacity using Alizarin red staining after 28 days of hOBs culture ($n = 6$). These cells were transfected with both the mimic and inhibitor of miR-320a and the corresponding controls at day 1 and day 14 after seeding. In the case of osteosarcoma lines, MG-63 ($n = 3$) and U2OS ($n = 3$), cells were transfected at day 1 after seeding, and Alizarin red assays were performed after 7 days of culture.

For Alizarin red assays, wells were washed with PBS and fixed with 10% formalin for 10 min at room temperature. Then, cells were washed with PBS and stained with a 40 mM Alizarin red solution at pH 4.2 (Merck Life Science S.L.U., Madrid, Spain) for 10 min under gentle shaking. After that, samples were washed carefully with PBS to remove excess stain, and cell mineralization was quantified by dissolving the precipitated Alizarin red assays with a 10% cetylpyridinium chloride solution at room temperature during 30 min on gentle shaking. One hundred microliters of the stained solutions were measured with a scanning multiwell spectrophotometer using an absorbance at 550 nm.

## 2.7. ALP Activity Assay

Osteoblast differentiation was assessed by measuring ALP activity in hOBs ($n = 3$), MG-63 ($n = 3$), and U20S ($n = 3$) at 48 h after the transfection of miR-320a or its inhibitor using the Alkaline Phosphatase Assay Kit (Colorimetric) (Abcam; Cambridge, UK) according to manufacturer instructions.

## 2.8. Oxidative Stress Evaluation

CellRox® Green Reagent (10 μM) (Invitrogen; Carlsbad, California, USA) was used to assess the oxidative stress levels at 72 h after transfection by following manufacturer instructions. This reagent is used for the detection and quantitation of reactive oxygen species (ROS) in live cells. Fluorescence levels were evaluated through the LEICA DMIL LED fluorescence microscope using the Leica Application Suite (Leica Microsystems). ImageJ software was used for fluorescence comparisons among different transfection conditions.

## 2.9. Bioinformatic Analyses

Targets and pathways of miR-320a were predicted using the DIANA-miRPath v3. 0 (http://snf-515788.vm.okeanos.grnet.gr/#mirnas=hsa-miR-320a&methods=Tarbase&selection=0) , which is a computational tool [17] used to identify molecular pathways potentially altered by the

miRNA transfection. The platform miRNet (https://www.mirnet.ca/miRNet/home.xhtml) was used to construct the interaction network between miR-320a and target genes from data collected from four well-annotated databases, i.e., miRTarBase v7.0, TarBase v7.0, miRecords, and miRanda.

### 2.10. Statistical Analysis

The Mann–Whitney U test in the SPSS v.12.0 for Windows was used to compare miR-320a mimic and inhibitor with their respective controls. All analyses were two-tailed, and *p*-values of <0.05 were considered significant.

## 3. Results

### 3.1. miR-320a Levels Quantification

Basal expression levels of miR-320a were quantified by qPCR in hOB ($n = 2$), U2OS ($n = 2$) and MG-63 ($n = 2$) cells. Osteosarcoma cells showed lower miR-320a expression levels compared to hOB cells, mainly MG-63 cells (Figure 1*).

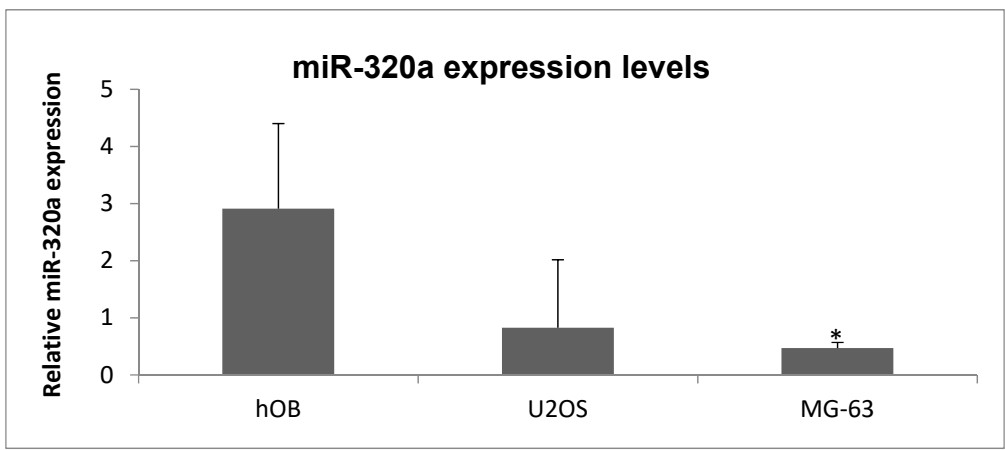

**Figure 1.** miR-320a expression levels assessed by qPCR in human osteoblast (hOB), U2OS, and MG-63 cells. Data represent the mean ± SD ($n = 2$). * $p < 0.05$ for hOB cells.

### 3.2. Cell Viability and Proliferation

The transfection of the miR-320a mimic caused a significant reduction in cell viability in MG-63 cells ($p = 0.046$) (Figure 2A). A similar trend was observed in U2OS viability after mimic transfection, while an increase of viability ($p = 0.045$) was observed upon the inhibition of miR-320a. In contrast, hOBs viability was not affected by miR-320a transfection.

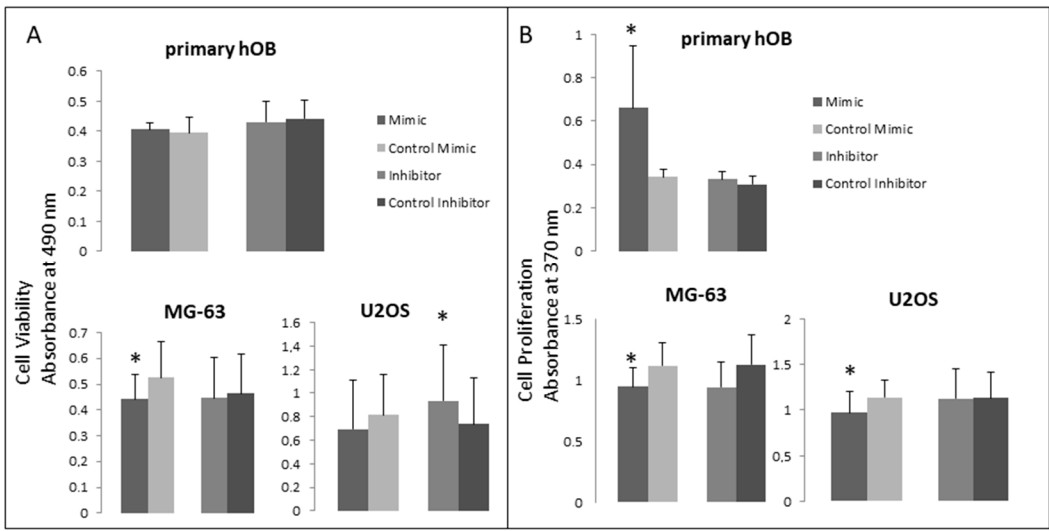

**Figure 2.** Effects of miR-320a on cell viability (**A**) and proliferation (**B**) in primary hOBs (*n* = 3), MG-63 (*n* = 3), and U2OS cells (*n* = 3) at 48 h post-transfection. Data represent the mean ± SD. * *p* < 0.05.

The overexpression of miR-320a significantly decreased MG-63 and U2OS proliferation (*p* = 0.028 and 0.04, respectively), which correlated with the viability results (Figure 2B). In contrast, the overexpression of miR-320a (*p* = 0.045) significantly increased hOB proliferation. In this case, the inhibitor transfection did not affect the proliferation in any cell type.

### 3.3. ALP Activity

A decrease of ALP activity was observed after mimic transfection in case of hOBs (*p* = 0.038). In contrast, the effects of ALP activity on MG-63 (*p* = 0.018) and U2OS cells (*p* = 0.028) were detected after inhibitor transfection, where ALP activity was increased after the inhibition of miR-320a (Figure 3).

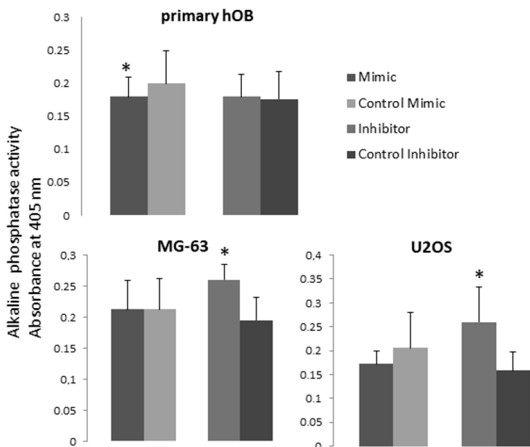

**Figure 3.** Effects of miR-320a on alkaline phosphatase activity in primary hOBs (*n* = 3), MG-63 (*n* = 3), and U2OS cells (*n* = 3) at 48 h post-transfection. Data represent the mean ± SD. * *p* < 0.05.

### 3.4. Assessment of Cell Mineralization Capacity

The overexpression of miR-320a reduced matrix mineralization in all cell lines tested, reaching significance in hOBs ($p = 0.02$) and U2OS cells ($p = 0.028$) (Figure 4). Inhibitor transfection did not affect significantly Alizarin red staining.

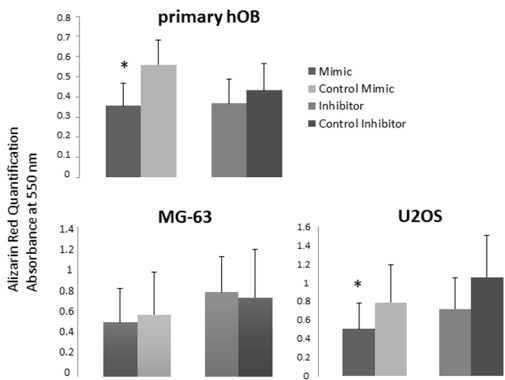

**Figure 4.** Effects of miR-320a transfection on the matrix mineralization of hOB and osteosarcoma cells. Alizarin red quantification was assayed in hOB cells ($n = 6$) after 28 days from transfection and in MG-63 ($n = 3$) and U2OS ($n = 3$) cells after 7 days from transfection. Data represent the mean ± SD. * $p < 0.05$.

### 3.5. Cellular Oxidative Stress Measurement

Osteosarcoma cells had higher ROS levels than hOBs at baseline (transfected with microRNA controls) (Figure 5). miR-320a overexpression increased the cell oxidative stress in all cells analyzed. Meanwhile, cells transfected with the miR-320a inhibitor showed an opposite effect (Supplementary Table S1). Since primary osteoblasts had very low ROS levels at baseline, the effect of the miR-320a inhibitor was subtly detected.

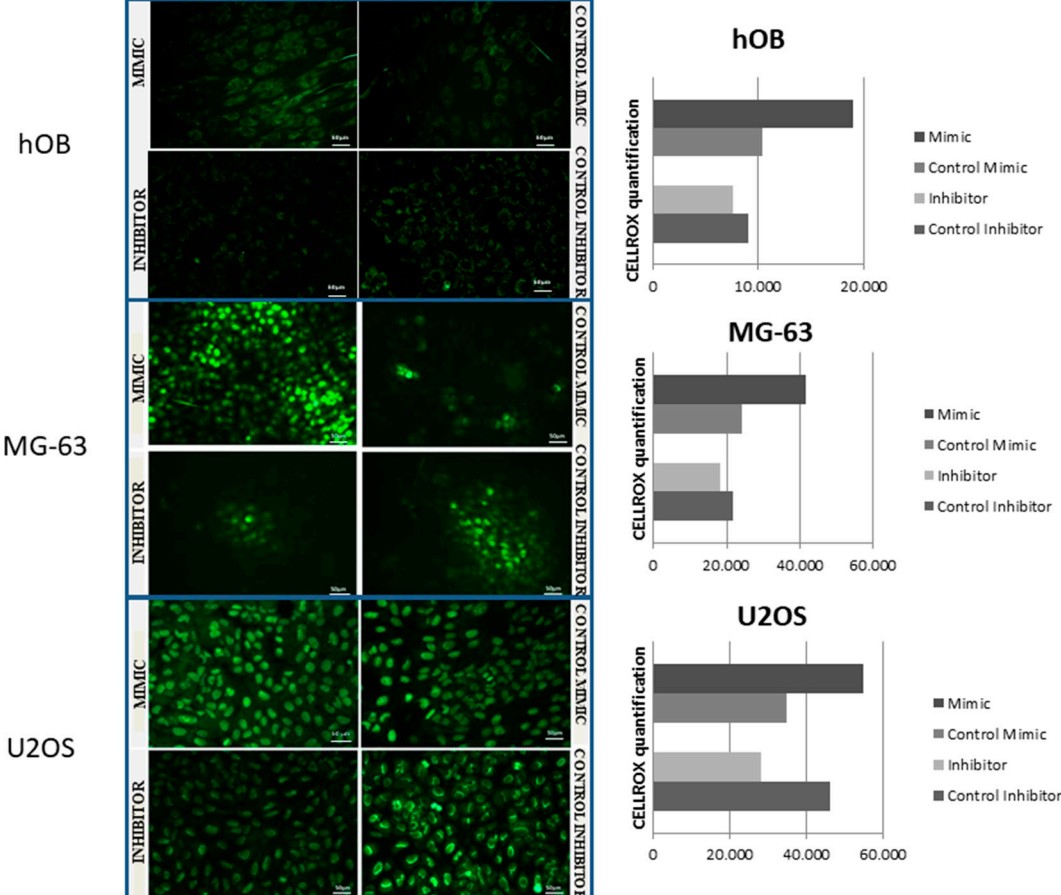

**Figure 5.** Effects of miR-320a on cell oxidative stress using CellRox® Green Reagent after 72 h of transfection. Images were obtained with the LEICA DMIL LED fluorescence microscope and with the Leica Application Suite. Fluorescence levels were quantified using the ImageJ software.

*3.6. Pathway Analysis*

In order to predict the overall effect of the miR-320a regulatory control, the miRPath 3.0 server platform was used to identify the cellular pathways potentially affected by this miRNA. Using the TarBase v7.0, which is based on experimentally supported miRNA–gene interactions, the most significant pathway was the TGF-beta signaling ($p$ = 1.32 × 10$^{-07}$) with 13 target genes including a number of genes of *SMAD* family, *BAMBI*, *MYC, MAPK1*, and *TGFBR2*. According to miRPath 3.0, these genes are involved in osteoblast differentiation, apoptosis, and cell cycle. In addition, the miRNet platform (Figure 6) showed pathways in cancer as a significantly enriched function ($p$ = 0.0209).

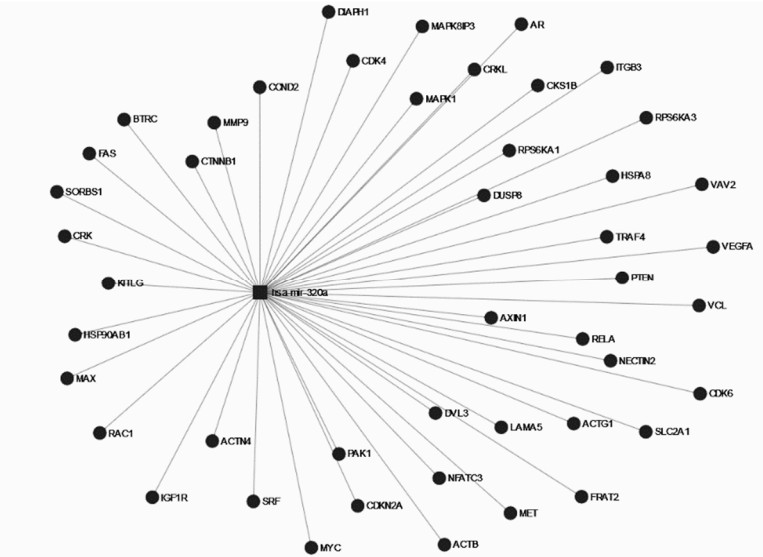

**Figure 6.** Network visualization of miR-320a targets enriched in cancer pathways using the miRNet server and the KEGG datasource.

## 4. Discussion

miR-320a was described as a tumor suppressor for osteosarcoma [9,11] and involved in osteoporotic bone [13], but its effect has never been compared between both cellular contexts. Hence, we overexpressed or inhibited miR-320a in two osteosarcoma cell lines, and the miRNA effects on these two cells lines were compared to that on primary hOBs. We observed an alteration in osteoblast function in all cells tested, involving reduced mineralization and increased oxidative stress. However, while viability and proliferation were impaired in osteosarcoma, no effect was detected in hOB viability and an increased proliferation was even observed. Interestingly, this miRNA is significantly downregulated in osteosarcoma tissues compared to in noncancerous tissues [9], and its overexpression significantly inhibited cell proliferation in U2OS [16]. In cervical cancer, miR-320a has also been found to be decreased and, similar to our study, to inhibit cell proliferation and to induce cell apoptosis [18]. In our study, this miRNA, at the tested doses, appeared to reduce the viability of osteosarcoma cells without affecting primary osteoblastic cells, suggesting that tumorigenic cells may be more sensitive to miR-320a toxicity. On the contrary, the overexpression of miR-320a in hOBs led to increased proliferation but a decreased mineralization capability and ALP activity, consistent with a loss of osteoblastic differentiation. It is in accordance with the increased levels found in fragility fracture [2], where high miR-320a levels may be involved in the osteoporotic phenotype. All these evidences suggested a different effect of miR-320a on tumorigenic cells compared to that on primary cells, where on one hand the viability and proliferation is reduced and on the other hand cells undergo reduced functionality. It is noteworthy that these osteosarcoma cells, which survived after miR-320a transfection, also showed an altered functioning in terms of osteoblastic performance. All these results strongly fit with an involvement of the TGF-beta signaling pathway predicted by DIANA-miRPath using miR-320a target genes with experimental support. The crucial role of TGF-beta signaling pathway in osteoblast proliferation, differentiation, and function is well-known [19]. In fact, TGF-beta/BMP signaling was proposed as an altered pathway in osteosarcoma cells, according to Yang et al. (2016) [20]. In that study, they analyzed transcriptional profiles of osteosarcomas, including two osteosarcoma biopsy specimens, two cell lines, and two xenografts derived from patient specimens–one from normal osteoblasts and one from mesenchymal stem cells. They concluded that alterations in the signaling axes of IGF-1 and TGF-beta, in concert with cell cycle regulators, could be involved in the pathogenic basis of osteosarcoma. However, of the 13 miR-320a target genes predicted in the TGF-beta pathway, only SMAD5 was validated in an RNA microarray analysis [13] after miR-320a mimic or inhibitor transfection (Supplementary Table S2). On the other hand, several genes predicted in cancer

pathways have been validated in this microarray (Supplementary Table S3). Since cellular oxidative stress can affect cell viability and function, we measured ROS in hOBs and osteosarcoma cells. The results showed that the overexpression of miR-320a produced increased ROS levels whereas miR-320a inhibition reduced ROS levels. In primary cells, with a normal functioning, minimal ROS levels were observed, since they were processed by the cellular antioxidant defense system. The increased ROS levels due to miR-320a overexpression could be involved in part in the altered functionality [21] provoking cell proliferation increase [22]. The elevated ROS levels of osteosarcoma cells, compared to those of untransformed cells, may reflect mitochondrial dysfunction and an accelerated metabolism [23]. However, cancer cells are more sensitive than untransformed cells to an acute increase in ROS levels. Therefore, an overproduction of intracellular ROS can induce cancer cell cycle arrest, senescence, and apoptosis [24]. The increasing levels of ROS due to miR-320a overexpression shown in our study would provoke the observed decrease in osteosarcoma cell viability without affecting the viability of primary osteoblasts.

It is noteworthy that microRNA studies performed on established cell lines provide data valid only for these cells. Therefore, the results obtained cannot be extrapolated to normal osteoblasts and normal bone tissue.

A limitation of the study is the different effects detected between the mimic and the inhibitor, possibly due to the cellular levels of miR-320a, since the inhibitor effect would be conditioned to the amount of miRNA present in the cells. Thus, if the amount of miRNA is very small, its action will be better observed after mimic transfection rather than after inhibitor transfection. If miRNA expression is very high, its action will be better observed after inhibitor transfection rather than after mimic transfection. This could explain why in some kinds of cells we observed the mimic effect whereas in other cells the effect would be observed through the inhibitor.

Moreover, the miR-320a effect should be corroborated using xenograft models, through the transplantation of MG-63 and U2OS cells, in order to estimate tumor suppressor effects.

In summary, the overexpression of miR-320a produces an increase in oxidative stress levels and a reduced functionality in both hOB and osteosarcoma cells. On the other hand, osteosarcoma cells showed a decreased viability and proliferation unlike primary cells that expressed a dedifferentiated phenotype.

**Supplementary Materials:** The following are available online at www.mdpi.com/2076-3417/10/8/2852/s1, Table S1: Measurement of fluorescence levels of CellRox Green Reagent Assay (ImageJ platform); Table S2: Effect of miRNA-320a on predicted targeted genes in "TGF-beta signaling pathway"; Table S3: Effect of miRNA-320a on predicted targeted genes in "Pathways in cancer". Only significant genes differentially expressed in microarray analysis.

**Author Contributions:** Conceptualization, L.D.-U., S.B., D.G., A.D.-P., and N.G.-G.; data curation, L.D.-U.; formal analysis, D.G. and N.G.-G.; funding acquisition, L.D.-U., R.G.-F., A.D.-P., and X.N.; investigation, L.D.-U. and N.G.-G.; methodology, L.D.-U. and N.G.-G.; project administration, X.N.; supervision, A.D.-P. and N.G.-G.; validation, L.D.-U., S.B., and D.G.; writing of the original draft, S.B., D.G., and N.G.-G.; writing of review and editing, L.D.-U., S.B., R.G.-F., D.G., A.D.-P., X.N., and N.G.-G.

**Funding:** This research was supported by the CIBER on Frailty and Healthy Ageing (CIBERFES; grant number: CB16/10/00245), the CIBERER (grant number: U720), FEDER funds, and grants from the Science and Innovation Ministry (ISCIII; grant numbers: PI16/01860 and PI13/00116; SAF2016-75948-R). L.D.-U was granted with a PFIS predoctoral fellowship from the ISCIII.

**Conflicts of Interest:** The authors declare no conflicts of interest.

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
