# Peer review of "Effect of the Tumor Suppressor miR-320a on Viability and Functionality of Human Osteosarcoma Cell Lines Compared to Primary Osteoblasts"

_applsci, doi:10.3390/app10082852_

Round 1

Reviewer 1 Report

In their manuscript “Effect of the tumor suppressor miR-320a on viability and functionality of human osteosarcoma cell lines compared to primary osteoblasts”, De-Ugarte and colleagues report on the results of several in vitro cell assays (cell viability, proliferation, oxidative stress, matrix mineralization, alkaline phosphatase activity) in the human osteosarcoma cell lines MG-63 and U2OS as well as in human primary osteoblasts (hOBs) from prosthesis replacement after miR-320a overexpression or silencing by mimics or inhibitors, respectively. The authors show that overexpression of miR-320a in both osteosarcoma cells and hOBs reduced mineralization capacity and increased oxidative stress could be detected in all cells after miR-320a overexpression.

Comments:

+ As one of the rationales for their study, the authors state that while miR-320a has been extensively studied in osteosarcoma and primary osteoblasts in a separated manner, it had never been tested in both cell types in the same experimental context. It was not clear to me at all what exactly the benefit of this study is compared to already known partial data like that overexpression of miR-320a significantly inhibited cell proliferation in MG-63 and U2OS MG-63 cells (Wu et al., J Cancer Res Ther 2016). The present manuscript looks to me a lot like the publication in PLOS ONE from the same group in 2018 (which is of course nothing "bad"), which also has a strong methodological overlap, except that no data in MG-63 and U2OS cells are presented there.

+ Please provide data analyzing the extent of miR-320a overexpression/inhibition in cells/cell lines compared to the control transfections (e.g., by qRT-PCR), so the readers of the publication are better able to judge about the effects observed.

+ miRNAs almost always give rise to two strands (-3p, -5p), the ratio of which can be quite different depending on e.g. the organ/tissue/cell type analyzed. The authors should clarify which strand of miR-320a they were analyzing in their study.

+ All assays were performed 48 hours after transfection, only the oxidative stress evaluation was performed after 72 hours. What is the reason for this ?

+ The authors state on p.3/line 119 that target genes from four databases were used, yet they provide only three names. Is the fourth one missing or were only three databases used ?

+ Please change decimal signs in figures from “,” to “.”

+ The labeling within Fig. 4 (inhibitor, mimic, control inhibitor, control mimic) is not readable at all. Please increase font size or arrange pictures differently.

+ The “author contribution” on p.7/line 227 still contains the journal instructions for funding enumeration. Please remove.

Author Response

Dear Reviewer,

We truly appreciate all the constructive comments and suggestions that have improved our manuscript.

We are grateful for the opportunity to amend our manuscript with the changes suggested. The revisions are included in the attached manuscript; new sentences and other changes in the manuscript are highlighted in yellow. Our answers are detailed below these lines.

We thank you in advance for your attention to our work.

Reviewer’s comments

1). As one of the rationales for their study, the authors state that while miR-320a has been extensively studied in osteosarcoma and primary osteoblasts in a separated manner, it had never been tested in both cell types in the same experimental context. It was not clear to me at all what exactly the benefit of this study is compared to already known partial data like that overexpression of miR-320a significantly inhibited cell proliferation in MG-63 and U2OS MG-63 cells (Wu et al., J Cancer Res Ther 2016). The present manuscript looks to me a lot like the publication in PLOS ONE from the same group in 2018 (which is of course nothing "bad"), which also has a strong methodological overlap, except that no data in MG-63 and U2OS cells are presented there.

We totally agree with the reviewer that this article is an extension of Wu et al. and from an own article. However, we believe that this article provides new data in two points:

  1. a) Respect to Wu et al.: they tested proliferation in U2OS giving similar results to our study but OB function was not evaluated. We included in our study other functional assessments including mineralization capacity as well as ROS levels. Besides, Wu et al. provided valuable information about miR-320a expression on different osteoblastic cell lines and normal osteoblasts which was a starting point for our study.
  2. b) Most of the microRNAs studies have been performed on stablished cell lines surely due to the difficulties for obtaining primary osteoblasts. Here, we present results comparing normal osteoblasts and osteosarcoma cells which suggest that the behavior of both cell types can differ in some aspects and therefore, results obtained in cell lines cannot be extrapolated to normal osteoblasts and normal bone tissue.

From our previous study published in PLOS ONE, we hypothesized that the oxidative status of the cells can modulate the effect of miRNA-320a since this is involved in ROS levels, and ROS levels can affect cell viability and function. Here, we found increased oxidative stress in osteosarcoma cells, respect to normal cells, and miR-320a even more increases these levels.

In order to clarify these points, we have included several comments in introduction and discussion.

Introduction: In a previous study by Wu et al. (2016) [16], a reduced miR-320a expression was observed in several osteosarcoma cell lines compared to human normal osteoblastic cell line hFOB1.19, but the miRNA-320a effect on cell proliferation only was tested in U2OS.

Discussion: It is noteworthy that microRNA studies performed on established cell lines provide data valid only for these cells and therefore, the results obtained cannot be extrapolated to normal osteoblasts and normal bone tissue.

2) Please provide data analyzing the extent of miR-320a overexpression/inhibition in cells/cell lines compared to the control transfections (e.g., by qRT-PCR), so the readers of the publication are better able to judge about the effects observed.

These data were published in our previous study. According to reviewer suggestion we have included it in methods:

Transfection efficiency was controlled using qPCR and miRIDIAN microRNA Mimic Transfection Control with Dy547 as previously described in De-Ugarte et al. [13]. The mature miR-320a sequence corresponds to hsa-miR-320a-3p (5’-AA AAGCTGGGTTGAGAGGGCGA-3’).  Same efficiencies were detected in all tested cell lines (data not shown).

3) + miRNAs almost always give rise to two strands (-3p, -5p), the ratio of which can be quite different depending on e.g. the organ/tissue/cell type analyzed. The authors should clarify which strand of miR-320a they were analyzing in their study.

Thanks for the suggestion. We have included this information in methods

The mature miR-320a sequence corresponds to hsa-miR-320a-3p (5’-AA AAGCTGGGTTGAGAGGGCGA-3’). 

4) All assays were performed 48 hours after transfection, only the oxidative stress evaluation was performed after 72 hours. What is the reason for this ?

Mainly due to the low ROS levels detected in primary cells. The highest effect of miR-320a on oxidative stress was observed after 72 hours.

5) The authors state on p.3/line 119 that target genes from four databases were used, yet they provide only three names. Is the fourth one missing or were only three databases used ?

Sorry, it was a mistake. We have amended the sentence:

The platform miRNet (https://www.mirnet.ca/miRNet/home.xhtml) was used to construct the interaction network between miR-320a and target genes from data collected from four well-annotated database miRTarBase v7.0, TarBase v7.0 , miRecords and miRanda.

6) Please change decimal signs in figures from “,” to “.”

Thanks for the appreciation. Figures are corrected.

7) The labeling within Fig. 4 (inhibitor, mimic, control inhibitor, control mimic) is not readable at all. Please increase font size or arrange pictures differently.

We have improved the figure.

8) The “author contribution” on p.7/line 227 still contains the journal instructions for funding enumeration. Please remove.

Thanks for the comment. It has been amended.

Reviewer 2 Report

In this paper, the authors estimated that one of miRNAs, miR-320a, involves in cell viability, proliferation, and oxidative stress, matrix mineralization, and alkaline phosphatase (ALP) activity in a separated manner between osteosarcoma cell lines and primary human osteoblasts. This paper is a report that miR-320a and subsequent signaling may serve to facilitate biological features of osteoblasts and osteosarcoma cells and showed significant data in molecular oncology. However there are some defects in data and descriptions. Criticisms regarding this revised paper are discussed below.

Comments

1. Reliability of experiments miR-320a overexpression or inhibition using mirVanaTM miRNA mimic or inhibitor.                                                          There are no quantitative data of original expressions of miR-320a in MG-63 and U2OS cells and hOB cells and altered expressions by mirVanaTM miRNA mimic or inhibitor. Please show these data using RT-qPCR for miRNA etc.

2. Differential pathway between osteosarcoma cells and normal osteoblasts

a) The authors mentioned TGF-beta signaling with13 target genes involved by miR-320a using the pathway analysis. Please reveal altered expressions of SMADs mRNA or protein in these cells of overexpressed / inhibited miR320a

b) Is there a difference between osteosarcoma cells and normal osteoblasts in TGF-beta family signaling?

3. In vivo tumorigenesis of MG-63 and U2OS cells with altered miR-320a expressions in xenograft models.                                                                The experiments of xenograft models should be done using transplantation of MG-63 and U2OS cells with overexpressed / inhibited miR-320a into immunodeficient mice in order to estimate tumor suppressed effects.

Author Response

Dear reviewer,

We truly appreciate all the constructive comments and suggestions that have improved our manuscript.

We are grateful for the opportunity to amend our manuscript with the changes suggested. The revisions are included in the attached manuscript; new sentences and other changes in the manuscript are highlighted in yellow. Our answers are detailed below these lines.

We thank you in advance for your attention to our work.

Reviewer’s Comments

  1. Reliability of experiments miR-320a overexpression or inhibition using mirVanaTM miRNA mimic or inhibitor.                                                          There are no quantitative data of original expressions of miR-320a in MG-63 and U2OS cells and hOB cells and altered expressions by mirVanaTM miRNA mimic or inhibitor. Please show these data using RT-qPCR for miRNA etc.

Thanks for the comment. Actually, all these data are previously published by Wu et al., J Cancer Res Ther 2016 and an own previous study. We have added this information in the introduction and methods.

Introduction: In a previous study by Wu et al. (2016) [16], it was observed a reduced miR-320a expression in several osteosarcoma cell lines compared to human normal osteoblastic cell line hFOB1.19, but the miRNA-320a effect on cell proliferation was only tested in U2OS.

Methods: Transfection efficiency was controlled using qPCR and miRIDIAN microRNA Mimic Transfection Control with Dy547 as previously described in De-Ugarte et al. [13]. The mature miR-320a sequence corresponds to hsa-miR-320a-3p (5’-AAAAGCTGGGTTGAGAGGGCGA-3’).  Same efficiencies were detected in all tested cell lines (data not shown).

Actually, we used the observed results from Wu et al. as a starting point for our study. See figure below these lines:

  1. Differential pathway between osteosarcoma cells and normal osteoblasts
  2. a) The authors mentioned TGF-beta signaling with13 target genes involved by miR-320a using the pathway analysis. Please reveal altered expressions of SMADs mRNA or protein in these cells of overexpressed / inhibited miR320a

We previously performed a microarray analysis, published in De-Ugarte et al. (2018) [13], after miR-320a mimic or inhibitor transfection in 5 human osteoblasts lines (available in GEO: GSE121892). According to reviewer suggestion, we have now analyzed these microarray results exploring expression differences in genes of TGF-beta signaling and Cancer signaling from the in silico analysis. We have added these data in discussion and supplemental table 2 and 3.

However, of the 13 miR-320a target genes, predicted in the TGF-beta pathway, only SMAD5 was validated in a microarray analysis [13] after miR-320a mimic or inhibitor transfection (supplemental table 2). On the other hand, several genes predicted in cancer pathways have been validated in this microarray (supplemental table 3).

  1. b) Is there a difference between osteosarcoma cells and normal osteoblasts in TGF-beta family signaling?

Indeed, TGF-beta/BMP signaling is proposed as an altered pathway in osteosarcoma cells according to Yang et al. Clin Orthop Relat Res 2016. They analyzed transcriptional profiles of osteosarcomas, including two primary biopsy specimens, two cell lines, two xenografts derived from patient specimens, and one from normal osteoblasts and from mesenchymal stem cells and they concluded that the alterations in the signaling axes of IGF-1 and TGF-b, in concert with cell cycle regulators, may be an important pathogenic basis of osteosarcoma.

Moreover, we found in our microarray a significant alteration of BMP2, PTGS2, and IGFBP1 genes corroborating that these pathways could be affected by miR-320a.

We have added this information in the discussion.

In fact, TGF-beta/BMP signaling is proposed as an altered pathway in osteosarcoma cells according to Yang et al. (2016) [20]. In this study, they analyzed transcriptional profiles of osteosarcomas, including two primary biopsy specimens, two cell lines, two xenografts derived from patient specimens, and one from normal osteoblasts and from mesenchymal stem cells and they concluded that alterations in the signaling axes of IGF-1 and TGF-beta, in concert with cell cycle regulators, could be involved in the pathogenic basis of osteosarcoma.

  1. In vivo tumorigenesis of MG-63 and U2OS cells with altered miR-320a expressions in xenograft models.      The experiments of xenograft models should be done using transplantation of MG-63 and U2OS cells with overexpressed / inhibited miR-320a into immunodeficient mice in order to estimate tumor suppressed effects.

We totally agree with the reviewer that our results should be demonstrated using xenograft models. Unfortunately, we do not have the methodology neither our lab available to perform these experiments at this moment. We add it in limitations.

Moreover, the miR-320a effect should be corroborated using xenograft models using transplantation of MG-63 and U2OS cells in order to estimate tumor suppressor effects.

Reviewer 3 Report

Authors of this manuscript attemoted to address the role of the micro RNA miR-320a by parallel testing of primary osteoblasts and osteosarcoma cell lines. Though the effects of the micro RNA were previosly described, a thorough parallel testing of normal and cancerous cells was done performed previously. The authors used state of the art methods to investigate cell proliferation, viability and physiological function (mineralization) of this RNA species. Unfortunately, only minor differences were discovered, though some reached the level of statistical significance. Nevertheless, though potentially interesting, one may not be certain whether the findings described in the manuscript are of real (patho)physiological significance. 

In the introductory section the authors draw attention to the possible involvement of miR-320a in osteoporosis and osteonecrosis of the femoral head. It is disappointing that these clinically important abnormalities or other clinically relevant questions (eg osteosarcoma growth or response to certain therapy) are not really investigated using the test system that is obviously well working iin their hands. Therefore this nice paper - in my opinion - remains a technical one.

Author Response

Dear reviewer,

We truly appreciate all the constructive comments and suggestions that have improved our manuscript.

We are grateful for the opportunity to amend our manuscript with the changes suggested. The revisions are included in the attached manuscript; new sentences and other changes in the manuscript are highlighted in yellow. Our answers are detailed below these lines.

We thank you in advance for your attention to our work.

Reviewer’s Comments

1)Authors of this manuscript attempted to address the role of the micro RNA miR-320a by parallel testing of primary osteoblasts and osteosarcoma cell lines. Though the effects of the micro RNA were previously described, a thorough parallel testing of normal and cancerous cells was done performed previously. The authors used state of the art methods to investigate cell proliferation, viability and physiological function (mineralization) of this RNA species. Unfortunately, only minor differences were discovered, though some reached the level of statistical significance. Nevertheless, though potentially interesting, one may not be certain whether the findings described in the manuscript are of real (patho)physiological significance. In the introductory section the authors draw attention to the possible involvement of miR-320a in osteoporosis and osteonecrosis of the femoral head. It is disappointing that these clinically important abnormalities or other clinically relevant questions (eg osteosarcoma growth or response to certain therapy) are not really investigated using the test system that is obviously well working iin their hands. Therefore this nice paper - in my opinion - remains a technical one.

We totally agree with the reviewer that is not clear whether the findings described in the manuscript are of real (patho)physiological significance. We really need more experiments to demonstrate it. Unfortunately, we do not have the methodology neither our lab available to perform these experiments at this moment.

In this regard, we have included a statement into the limitations section:

Moreover, the miR-320a effect should be corroborated using xenograft models using transplantation of MG-63 and U2OS cells in order to estimate tumor suppressor effects.

Our work intended to contribute in some points:

  1. a) A previous study by Wu et al. J Cancer Res Ther 2016 provided valuable information about miR-320a expression on different osteoblastic cell lines and normal osteoblasts which was a starting point for our study. They observed lower miR-320a expression in osteosarcoma cells respect to normal osteoblastic cells. Indeed, Wu et al. tested the effect of miR-320a on proliferation in U2OS giving similar results to our study, but OB function was not evaluated. We included in our study other functional assessments including mineralization capacity as well as ROS levels.

Moreover, from our previous study published in PLOS ONE (De-Ugarte et al. (2018)), we hypothesized that the oxidative status of the cells can modulate the effect of miRNA-320a since this is involved in ROS levels, and ROS levels can affect cell viability and function. Here, we found increased oxidative stress in osteosarcoma cells, respect to normal cells, and miR-320a even more increases these levels.

  1. b) Most of the microRNAs studies have been performed on stablished cell lines surely due to the difficulties for obtaining primary osteoblasts. Here, we present results comparing normal osteoblasts and osteosarcoma cells which suggest that the behavior of both cell types can differ in some aspects and therefore, results obtained in cell lines cannot be extrapolated to normal osteoblasts and normal bone tissue.

In order to clarify these points, we have included several comments in introduction and discussion.

Introduction: In a previous study by Wu et al. (2016) [16], a reduced miR-320a expression was observed in several osteosarcoma cell lines compared to human normal osteoblastic cell line hFOB1.19, but the miRNA-320a effect on cell proliferation only was tested in U2OS. Moreover, no other OB functional assessments were evaluated.

Discussion: It is noteworthy that microRNA studies performed on established cell lines provide data valid only for these cells and therefore, the results obtained cannot be extrapolated to normal osteoblasts and normal bone tissue.

Additionally, new data have been included in the discussion section:

In fact, TGF-beta/BMP signaling is proposed as an altered pathway in osteosarcoma cells, according to Yang et al. (2016) [20]. In that study, they analyzed transcriptional profiles of osteosarcomas, including two primary biopsy specimens, two cell lines, two xenografts derived from patient specimens, and one from normal osteoblasts and from mesenchymal stem cells and they concluded that alterations in the signaling axes of IGF-1 and TGF-beta, in concert with cell cycle regulators, could be involved in the pathogenic basis of osteosarcoma. However, of the 13 miR-320a target genes predicted in the TGF-beta pathway, only SMAD5 was validated in a RNA microarray analysis [13] after miR-320a mimic or inhibitor transfection (supplemental table 2). On the other hand, several genes predicted in cancer pathways have been validated in this microarray (supplemental table 3).

Round 2

Reviewer 2 Report

In this paper, the authors estimated that one of miRNAs, miR-320a, involves in cell viability, proliferation, and oxidative stress, matrix mineralization, and alkaline phosphatase (ALP) activity in a separated manner between osteosarcoma cell lines and primary human osteoblasts. This paper is a report that miR-320a and subsequent signaling may serve to facilitate biological features of osteoblasts and osteosarcoma cells and showed significant data in molecular oncology. In the revised paper, the authors improved in certain descriptions at several questionable points. However there are some defects in data and descriptions. Criticisms regarding this revised paper are discussed below.

Comments in the first review : Reliability of experiments miR-320a overexpression or inhibition using mirVanaTM miRNA mimic or inhibitor. There are no quantitative data of original expressions of miR-320a in MG-63 and U2OS cells and hOB cells and altered expressions by mirVanaTM miRNA mimic or inhibitor.

Comments for second review : The authors explained that “all these data are previously published by Wu et al., J Cancer Res Ther 2016 and an own previous study” about this point. Wu et al. used human osteoblast cellline, hFOB1.19, as human normal osteoblast, however the authors had experiments using human primary osteoblasts. Precisely these data, such as Fig 1,2,3, and 4, in this paper derived from all experiments which should be examined for this designed study newly. Please show these data using RT-qPCR for miRNA etc.

Author Response

We really thanks for the effort to improve our manuscript. The revisions are included in the attached manuscript; new sentences and other changes in the manuscript are highlighted in yellow. Our answers are detailed below these lines.

Reviewer’s comment

The authors explained that “all these data are previously published by Wu et al., J Cancer Res Ther 2016 and an own previous study” about this point. Wu et al. used human osteoblast cellline, hFOB1.19, as human normal osteoblast, however the authors had experiments using human primary osteoblasts. Precisely these data, such as Fig 1,2,3, and 4, in this paper derived from all experiments which should be examined for this designed study newly. Please show these data using RT-qPCR for miRNA etc.

According to reviewer’s suggestion, We have included a new figure (figure 1 in the new version) into the article with qPCR data on miR-320a expression at baseline (without mimic and inhibitor transfection). Therefore, we included a new section in results. Moreover, we have also included the qPCR methodology in the methods section.

In methods:

MiR-320a quantification by qPCR

To evaluate the miR-320a expression levels in hOBs, U2OS and MG-63, total RNA was extracted using the miRNeasy mini kit (Qiagen) according to manufacturer instructions. Then, 1 µg of total RNA was reverse-transcribed using the miScript II RT kit (Qiagen). cDNA was assayed in 10 µl qPCR reactions in 384-well plates using MiScript SYBR Green PCR kit according to the protocol. The mature miR-320a sequence was used as a forward primer and the Universal primer as a reverse. Amplification was performed in a QuantStudio 12K Flex Real-Time PCR (Applied Biosystems), and the ExpressionSuite software v.1.0.3 (Life Technologies) was used for determination of relative quantification (RQ) by 2-ΔΔCt method. Global normalization was used to normalize qPCR results.

In results:

MiR-320a levels quantification

Basal expression levels of the miR-320a were quantified by qPCR in hOB (n=2), U2OS (n=2) and MG-63 (n=2).  Osteosarcoma cells showed lower miR-320a expression levels compared to hOB cells, mainly MG-63 cells (Figure 1).

Reviewer 3 Report

Thanks for your response and improvement of the manuscript.

Author Response

we really thanks the comment of the reviewer